# TRUSTED MULTI-RATER SEGMENTATION

## ABSTRACT

Deep learning models have shown strong performance in medical image segmentation. But their integration into clinical practice has been slow, largely due to the lack of reliable uncertainty estimates. In medical imaging, uncertainty arises not only from the input data, but also from inter-rater variability in annotations. Most existing multi-rater segmentation approaches focus on modeling label disagreement through probabilistic outputs, without providing explicit uncertainty estimates. We propose Trusted Multi-Rater Segmentation (TMS), a novel algorithm that integrates evidential deep learning into multi-rater medical image segmentation. TMS treats network outputs associated with each annotator as subjective opinions, represented as parameters of the Dirichlet distribution, and combines them using weighted belief fusion from subjective logic. Unlike prior methods, TMS produces both probabilistic segmentations and explicit, interpretable uncertainty estimates. We demonstrate state-of-the-art performance in the optic disc and cup segmentation tasks using the RIGA dataset, as well as lung nodule segmentation using the LIDC dataset. Moreover, we go beyond conventional performance measures by explicitly evaluating the quality of uncertainty estimates, showing that TMS exhibits strong uncertainty-awareness.

## 1 INTRODUCTION

Recent advances in deep learning (DL) have led to impressive progress across various medical image segmentation tasks. However, integration into clinical practice remains limited. High performance alone is not enough to gain the trust of experts, especially in a safety-critical field like healthcare. Neural networks are often overconfident, producing incorrect predictions with high confidence (Guo et al., 2017; Nguyen et al., 2015). Meanwhile, providing a reliable uncertainty estimate would allow a network to signal possible errors in its predictions. As a result, there is growing interest in the uncertainty quantification (UQ) task, which aims to improve the reliability of a model by enabling it to identify challenging cases and to defer final decisions to human experts.

Uncertainty, according to its sources, is generally categorized into two main types (Der Kiureghian & Ditlevsen, 2009). Epistemic uncertainty (EU) arises from a lack of knowledge; thus, it can be reduced by increasing the amount of training data or by adopting a more suitable model architecture. Aleatoric uncertainty (AU) is associated with inherent randomness in the data and is irreducible. As mentioned by Lambert et al. (2024), in medical image analysis, AU may stem not only from input data, but also from manual annotations, due to inter-rater variability. This has led to the emergence of multi-rater image segmentation, where multiple, potentially conflicting, annotations are available for each image.

A common strategy in multi-rater image segmentation is to fuse the provided annotations into a single binary ground-truth mask, enabling the use of standard segmentation approaches. Popular fusion methods include majority voting and STAPLE (Warfield et al., 2004). However, as reported by Jungo et al. (2018), models trained on fused ground-truth masks exhibit overconfidence and tend to underestimate the uncertainty present in the original annotations.

Recently, there has been a growing interest in learning strategies for multi-rater image segmentation. These include label-sampling (Jungo et al., 2018), using separate decoders for each rater (Hu et al., 2023), as well as applying variational Bayesian inference methods (Kohl et al., 2018; Hu et al., 2019) which can generate an infinite number of plausible predictions. However, most existing approaches primarily focus on producing predictions that reflect inter-rater variability, while overlooking the

task of explicitly estimating uncertainty. This limitation reduces their practicality in clinical settings, where understanding model confidence is essential for informed decision-making.

In this paper, we propose a new multi-rater medical image segmentation algorithm that simultaneously captures inter-rater variability and provides reliable uncertainty estimates. Our approach is based on evidential deep learning (EDL) (Sensoy et al., 2018)—a deterministic UQ framework that models a distribution over categorical distributions. EDL enables the model to express a lack of evidence, or, in other words, to say "I do not know," when faced with unfamiliar data.

To capture rater-specific behavior, we allocate distinct output channels for each individual annotator. Within the EDL framework, these outputs are interpreted as subjective opinions, each expressing both a prediction and its associated uncertainty. These opinions are subsequently combined using the weighted belief fusion (WBF) principle from subjective logic (SL) (Jøsang, 2016), enabling an uncertainty-aware aggregation of multiple subjective opinions into a combined opinion.

The key contributions of this work are as follows. 1. We propose Trusted Multi-Rater Segmentation (TMS)—an intuitive and interpretable learning pipeline for multi-rater medical image segmentation that leverages EDL and SL. To our knowledge, our approach is the first to apply EDL in a multi-rater setting. 2. Our model produces probabilistic segmentation maps and simultaneously quantifies EU and AU in a single forward pass. 3. We achieve state-of-the-art (SOTA) results for the tasks of optic disc and cup segmentation on the RIGA dataset, and for lung nodule segmentation on the LIDC dataset. 4. We go beyond conventional performance measures by explicitly evaluating the estimated uncertainty values, providing a more comprehensive assessment of the reliability of the model.

## 2 RELATED WORK

### 2.1 EVIDENTIAL DEEP LEARNING FOR UNCERTAINTY QUANTIFICATION

While model output probabilities are often interpreted as confidence scores, they tend to be miscalibrated and may not reflect true likelihoods (Lambert et al., 2024). Gawlikowski et al. (2023) classify UQ approaches into four main types: single deterministic methods (Malinin & Gales, 2018; Sensoy et al., 2018; Raghu et al., 2019), Bayesian methods (Blundell et al., 2015; Gal & Ghahramani, 2016), ensemble methods (Lakshminarayanan et al., 2017), and test-time augmentation methods (Ayhan & Berens, 2018). These methods differ in terms of computational requirements, need for architectural modifications, and types of uncertainty captured.

EDL (Sensoy et al., 2018) is a deterministic UQ approach based on Dempster–Shafer theory (DST) of evidence (Dempster, 1968) and SL (Jøsang, 2016). The network predicts non-negative evidence values for each class, which are then used to form the parameters of a Dirichlet distribution over categorical outcomes. This enables the model to represent both class-wise belief and overall uncertainty, instead of producing only a point estimate as in softmax-based networks. Applications of EDL in medical image segmentation include work by Zou et al. (2022) and Huang et al. (2021).

EDL has been extended to multi-view and multi-modal settings. A pioneering work is Trusted Multi-View Classification (Han et al., 2021), where evidence collected from different sources is combined using Dempster's combination rule, also referred to as belief constraint fusion (BCF) in SL. However, BCF may not be well suited for scenarios where the sources provide strongly conflicting evidence, as illustrated by Zadeh's example (Zadeh, 1996). To address this, recent studies have explored alternative strategies for aggregating opinions from multiple sources (Liu et al., 2022; Xu et al., 2024; Bezirganyan et al., 2025).

While there has been growing interest in applying EDL to multi-view and multi-modal tasks, where conflict arises between input sources, we are not aware of prior work using EDL for multi-rater image segmentation, where the conflict lies in the ground truth (GT) due to differing expert opinions.

### 2.2 MULTI-RATER MEDICAL IMAGE SEGMENTATION

A commonly adopted strategy in multi-rater medical image segmentation is to fuse individual expert annotations into a single proxy GT mask. Popular fusion techniques include majority voting, intersection, union, and STAPLE (Warfield et al., 2004). However, models trained on such combined masks often fail to capture the ambiguity in expert annotations and provide overconfident predic-

tions (Jungo et al., 2018). Another application of traditional learning paradigms to multi-rater segmentation is label sampling, where one of the annotations is randomly sampled per iteration (Jungo et al., 2018; Jensen et al., 2019).

To better preserve the inter-rater variability, some approaches model each rater's annotations independently, training separate prediction heads or decoders per rater. For example, Hu et al. (2023) propose a Bayesian neural network architecture featuring a one-encoder-multi-decoder design. The rater-specific representation is enhanced by integrating an attention module into each decoder.

Recent methods explicitly model annotator expertise, disagreement, or bias. For instance, MR-Net (Ji et al., 2021) incorporates prior knowledge about annotator reliability and leverages regions of disagreement to improve performance. The Transformer-based Annotation Bias-aware (TAB) model (Liao et al., 2023) accounts for annotator preference and stochastic errors to predict both meta and annotator-specific segmentations.

Some approaches focus on modeling a distribution of plausible segmentations rather than producing a single deterministic output. Probabilistic U-Net (Kohl et al., 2018) combines the U-Net architecture (Ronneberger et al., 2015) with a conditional variational autoencoder, enabling the generation of diverse segmentation hypotheses by sampling from a learned latent space. PHiSeg (Baumgartner et al., 2019) is a hierarchical probabilistic method, where separate latent variables are used to model the segmentation at different resolutions, leading to greater diversity in the predictions. In addition to producing diverse samples, PHiSeg analyzes segmentation variability by computing pixel-wise expected cross-entropy between the mean prediction and individual samples.

While existing methods provide probabilistic segmentation maps reflecting inter-rater variability or model a distribution of plausible segmentations, explicit modeling of uncertainty—particularly with a clear distinction between AU and EU—remains largely unexplored in the context of multi-rater medical image segmentation. In our literature review, we identified only a few studies that attempt to address this aspect. For instance, Hu et al. (2019) leverage inter-rater variability as a "GT" for AU, and introduce variational dropout to capture EU. However, EU is assessed only qualitatively through visual inspection. Gao et al. (2023) propose a Mixture of Stochastic Experts (MoSE) model to capture multimodal AU. Each expert network models a distinct uncertainty mode, while a gating network predicts their relevance per input. While an explicit measure of EU is not provided, the authors assess its presence indirectly by measuring pixel-wise entropy and sample diversity across varying training set sizes. The minimal changes observed in these scores suggest that EU contributes negligibly to the overall predictive uncertainty.

## 3 METHODS

### 3.1 EVIDENTIAL DEEP LEARNING

EDL is grounded in DST (Dempster, 1968) and its formalization in SL (Jøsang, 2016). SL defines a **subjective opinion**[1] through belief masses $b_k$ for each of the $K$ classes and an overall uncertainty $u$, such that

$$u + \sum_{k=1}^{K} b_k = 1, \quad u \geq 0, \quad b_k \geq 0 \quad \text{for } k = 1, \ldots, K. \tag{1}$$

The belief mass $b_k$ is derived from the evidence $e_k \geq 0$ which reflects the amount of evidence supporting class $k$. The belief masses $b_k$ and the overall uncertainty $u$ are computed as

$$b_k = \frac{e_k}{S}, \quad u = \frac{K}{S}, \tag{2}$$

where $S = \sum_{k=1}^{K}(e_k + 1)$. Notably, as the amount of observed evidence increases, the overall uncertainty decreases correspondingly. A subjective opinion corresponds to a Dirichlet distribution with parameters $\alpha_k = e_k + 1$ for $k = 1, \ldots, K$. Thus, $S = \sum_{k=1}^{K} \alpha_k$ defined above is the strength of Dirichlet. The Dirichlet distribution is a probability density function (PDF) over the possible values

---

[1]Subjective opinions also include a *base rate* vector $\mathbf{a} = (a_1, \ldots, a_K)^T$, representing prior probabilities over classes. In the absence of prior knowledge (as is common in DL), a uniform base rate $a_k = 1/K$ is typically assumed. We omit base rates in our notation for simplicity.

of a probability mass function (PMF) $\mathbf{p}$ (Sensoy et al., 2018). Unlike traditional neural networks, which typically produce a single point estimate using a softmax layer, EDL models a distribution over the categorical outputs. To enable this, the final softmax layer is replaced with an activation function that maps the output logits to non-negative values. These outputs are interpreted as evidence values $e_k$ for class $k$, from which the parameters of the Dirichlet distribution $\alpha_k$ are derived. Once these are obtained, the predictive class probabilities can be estimated using the mean of Dirichlet:

$$p_k = \frac{\alpha_k}{S}. \tag{3}$$

The loss function typically consists of two components. The first is the expected cross-entropy loss, computed as the integral of the conventional cross-entropy loss over the Dirichlet distribution:

$$\mathcal{L}_{\text{ace}}(\boldsymbol{\alpha}_i) = \int \left[ \sum_{k=1}^{K} -y_{ik} \log(p_{ik}) \right] \frac{1}{B(\boldsymbol{\alpha}_i)} \prod_{k=1}^{K} p_{ik}^{\alpha_{ik}-1} \, \mathrm{d}\mathbf{p}_i = \sum_{k=1}^{K} y_{ik} \left( \psi(S_i) - \psi(\alpha_{ik}) \right), \tag{4}$$

where $\boldsymbol{\alpha}_i$ and $\mathbf{p}_i$ denote the Dirichlet parameters and the class assignment probabilities on a simplex, respectively, for sample $i$; $y_{ik}$ and $p_{ik}$ are the GT label and the predicted probability, respectively, for sample $i$ and class $k$; and $\psi$ denotes the digamma function. This loss component helps to ensure that the evidence collected for the correct class exceeds that of the incorrect classes. However, it does not guarantee that the evidence for the incorrect classes will be low, or more specifically, driven to zero (Han et al., 2021). Thus, the following Kullback–Leibler (KL) divergence is introduced as the second loss component:

$$KL\left[ D\left( \mathbf{p}_i | \tilde{\boldsymbol{\alpha}}_i \right) \| D\left( \mathbf{p}_i | \mathbf{1} \right) \right] = \log \left( \frac{\Gamma\left( \sum_{k=1}^{K} \tilde{\alpha}_{ik} \right)}{\Gamma(K) \prod_{k=1}^{K} \Gamma(\tilde{\alpha}_{ik})} \right) + \sum_{k=1}^{K} (\tilde{\alpha}_{ik} - 1) \left[ \psi(\tilde{\alpha}_{ik}) - \psi\left( \sum_{j=1}^{K} \tilde{\alpha}_{ij} \right) \right], \tag{5}$$

where $\Gamma$ is the gamma function, and $\tilde{\boldsymbol{\alpha}}_i = \mathbf{y}_i + (1 - \mathbf{y}_i) \odot \boldsymbol{\alpha}_i$ is the adjusted parameter of the Dirichlet distribution, which is designed to prevent the evidence for the GT class from being reduced to zero. The overall loss is given as

$$\mathcal{L}(\boldsymbol{\alpha}_i) = \mathcal{L}_{\text{ace}}(\boldsymbol{\alpha}_i) + \lambda_t KL\left[ D\left( \mathbf{p}_i | \tilde{\boldsymbol{\alpha}}_i \right) \| D\left( \mathbf{p}_i | \mathbf{1} \right) \right], \tag{6}$$

where $\lambda_t$ is a balance factor. During training, $\lambda_t$ can be gradually increased to ensure the network does not prioritize the KL divergence early on (Han et al., 2021).

In EDL, EU can be quantified using $K/S$ from Equation 2 (Sensoy et al., 2018). The idea is that a larger denominator corresponds to higher confidence and thus lower uncertainty. AU can be quantified using the expected entropy of the data distribution $p(y|\mathbf{p})$. A low entropy suggests the model assigns high probability to a single class, whereas a high entropy implies a more spread-out distribution. When modeling the predictive distribution using a Dirichlet distribution, this expected entropy can be computed in closed form. Further details regarding the estimation of uncertainty in EDL are given by Ulmer et al. (2021).

## 3.2 PROPOSED APPROACH

We propose TMS, a novel multi-rater medical image segmentation network based on EDL. The pipeline of our method is visualized in Figure 1. We model the segmentation task as a multi-label problem, where the model predicts, for each pixel, which annotators would label it as foreground. TMS is built upon an encoder–decoder segmentation backbone, with the output layer producing $2R$ channels—foreground and background logits for each of the $R$ annotators. These logits are turned into non-negative evidence values via a Softplus activation, which are then transformed to parameterize a separate Dirichlet distribution for each rater. Thus, the network predictions corresponding to different annotators are treated as subjective opinions. To obtain a final prediction, the individual opinions are fused into a single opinion—equivalently, a combined Dirichlet distribution—that reflects the aggregated belief across annotators. It is important to note that our approach assumes each image is annotated by the same set of annotators consistently.

To perform the fusion, we draw on the concept of *belief fusion* from SL, which provides a principled framework for combining multiple subjective opinions into a unified one. Among the available

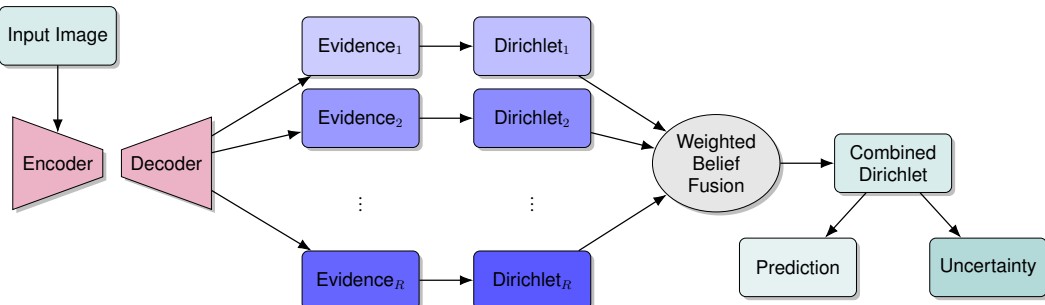

Figure 1: Visualization of TMS architecture. A shared encoder–decoder backbone produces rater-specific evidence maps, which are transformed into Dirichlet distributions representing subjective opinions. These opinions are combined using weighted belief fusion to form a combined Dirichlet distribution, from which predictions and uncertainty estimates are derived.

fusion strategies, we adopt *weighted belief fusion*, which is recommended in SL for scenarios in which multiple medical experts provide multinomial opinions over a shared set of diagnoses (Jøsang, 2016). To our knowledge, this is the first application of this fusion method within a DL framework, applied to network-derived segmentation outputs rather than to human expert opinions. Since the WBF operation is defined for two sources by Jøsang (2016) and is not associative, we employ the generalized version of this fusion for multiple sources introduced by Van Der Heijden et al. (2018). Formally, if, for a fixed pixel, $e_k^r$ represents the evidence collected for class $k$ by rater $r$, and $u^r$ is the associated uncertainty, then the fused opinion will correspond to the following evidence:

$$e_k = \frac{\sum_{r=1}^{R} e_k^r (1 - u^r)}{\sum_{r=1}^{R} (1 - u^r)}. \tag{7}$$

We compare WBF with two alternative fusion approaches from SL—*averaging belief fusion (ABF)* and *belief constraint fusion*. The generalized version of ABF for multiple sources, as described by Wang et al. (2017), is equivalent to averaging the evidence parameters obtained from each source:

$$e_k = \frac{\sum_{r=1}^{R} e_k^r}{R}. \tag{8}$$

WBF assigns weights to opinions based on their associated confidence. If all opinions are equally confident, the result is effectively an average. When one opinion is more confident and another more uncertain, the confident opinion has more weight. Not only does WBF address the limitation of treating all raters equally by weighting their opinions according to uncertainty, it also leverages the correlation between uncertainty and low-quality predictions. This enables performance gains even when the GT does not distinguish between raters. In comparison, a limitation of ABF, as pointed out by Bezirganyan et al. (2025), is that even under strong conflict between sources, uncertainty does not increase, making decisions derived from conflicting evidence appear as reliable as those from full agreement. BCF is included in the comparison due to its prior use in multi-view classification (Han et al., 2021), although its suitability for scenarios with conflicting beliefs has been questioned (Jøsang, 2016; Zadeh, 1996).

For the loss function, we follow Zou et al. (2022), where the authors leverage the EDL framework for brain tumor segmentation. The expected cross-entropy loss and the KL divergence term are applied independently for each pixel. To accommodate the nature of segmentation tasks, Dice loss is introduced as a third component, defined as

$$\mathcal{L}_{\text{Dice}} = 1 - \frac{2 y_{ik} p_{ik} + \alpha}{y_{ik} + p_{ik} + \beta}, \tag{9}$$

where $y_{ik}$ and $p_{ik}$ are the GT label and the predicted probability for sample $i$ and class $k$, respectively, and $\alpha$ and $\beta$ are smoothing factors. Thus, the loss function associated with rater $r$ is

$$\mathcal{L}^r = \frac{1}{N} \sum_{i=1}^{N} \left( \lambda_t \mathcal{L}_{\text{ace}}(\boldsymbol{\alpha}_i^r) + \lambda_s KL \left[ D \left( \mathbf{p}_i^r | \tilde{\boldsymbol{\alpha}}_i^r \right) \| D \left( \mathbf{p}_i^r | \mathbf{1} \right) \right] + \lambda_p \mathcal{L}_{\text{Dice}}^r, \tag{10}$$

where $N$ is the number of pixels. In our approach, we set $\lambda_t$ to 1, $\lambda_p$ to 0.1, and $\lambda_s$ gradually increases from 0 to 0.1. Our overall loss is given as

$$\mathcal{L}_{\text{overall}} = \sum_{r=1}^{R} \mathcal{L}^r. \tag{11}$$

While providing probability estimates that reflect the level of agreement between raters is essential, we argue that it is not sufficient for a comprehensive uncertainty analysis. Employing EDL allows us to complement the probabilistic segmentation maps with explicit uncertainty estimates, which enable us to assess the confidence of the network in its predictions.

Our method estimates epistemic and aleatoric uncertainty maps by applying the approaches mentioned in Section 3.1 at each pixel. However, a practical question when working with uncertainty is whether the overall prediction can be trusted (Lambert et al., 2024). While our method produces pixel-level uncertainty maps, it is crucial to aggregate these estimates into meaningful image-level scores. A commonly used aggregation strategy is to sum or average the pixel-level uncertainty estimates across the whole image. However, as pointed out by Kahl et al. (2024), for segmentation maps containing a single foreground object, the aggregated uncertainty score tends to correlate strongly with the size of the target object. To address this, we adopt the patch-level aggregation method described by Kahl et al. (2024), where a sliding window moves across the image, summing the uncertainties within each patch. The patch with the highest sum is selected, and the final image-level uncertainty score is obtained by averaging this sum over the patch size. We acknowledge that the optimal patch size is highly task-dependent and plan to explore more adaptive strategies in future.

## 4 DATASETS AND IMPLEMENTATION DETAILS

We evaluate our method on two publicly available multi-rater datasets. The first is RIGA (Almazroa et al., 2017), a retinal optic cup and disc dataset with annotations from six glaucoma experts, providing consistent rater identities across all images. The second is LIDC (Armato III et al., 2011), a widely used lung lesion segmentation dataset where each image is annotated by a subset of 4 out of 12 radiologists. Despite variability in annotator identity, we include it to enable meaningful comparisons with recent multi-rater segmentation methods, particularly those modeling uncertainty.

For the segmentation backbone, we employ the DeepLabV3 model (Chen et al., 2017) with a ResNet-101 backbone, initialized with pretrained weights. Training is performed using the Adam optimizer (Kingma & Ba, 2015), with 200 epochs for RIGA and 100 epochs for LIDC. The model with the lowest validation loss is selected for evaluation.

## 5 EVALUATION METHODS

Most existing work evaluates model predictions by measuring how well they reflect the agreement between annotators. To this end, we report the Soft Dice score, computed by averaging the Dice score over multiple probability thresholds. This captures how well the predicted probabilities align with the consensus across raters. In addition, some approaches focus on modeling a distribution of plausible segmentations, as discussed in Section 2.2. Such works commonly report the Generalized Energy Distance (GED) and Hungarian-Matched IoU (HM-IoU). Although generating multiple plausible segmentations is not an aim of our work, the fact that we model a distribution over categorical distributions allows us to sample different segmentation outputs. We therefore report GED and HM-IoU for comparison with prior work, since such comparisons are already limited by the lack of explicit uncertainty evaluation in most existing approaches.

To evaluate uncertainty estimates, we first assess whether the predicted aleatoric uncertainties reflect inter-rater variability using the Normalized Cross-Correlation (NCC), following Hu et al. (2019). NCC measures the similarity between predicted and reference uncertainty maps, normalized by their variance and means. The reference map is constructed by calculating the pixel-wise variance across rater annotations. To evaluate overall uncertainty estimates, we report the Area under the Referral Curve (AUCRef) described by Lambert et al. (2024), which measures how segmentation quality improves when the most uncertain predictions are progressively excluded, and the Area under the Risk-Coverage Curve (AURC) described by Kahl et al. (2024), which quantifies the trade-off

between minimizing risk and maximizing coverage. In both cases, pixel-level uncertainty estimates are aggregated to obtain a single uncertainty score per image, using the patch-level aggregation mechanism described in Section 3.2. Finally, we also include the Expected Calibration Error (ECE) following Gao et al. (2023) to quantify the calibration of predicted probabilities. Detailed definitions of all measures are provided in Appendix A.

# 6 RESULTS AND DISCUSSION

We first present results on the RIGA dataset, which serves as our primary benchmark due to its consistent rater identities. We compare our proposed TMS variants with two SOTA multi-rater segmentation approaches. MRNet (Ji et al., 2021) is widely used as a benchmark in prior work. TAB (Liao et al., 2023) represents a more recent SOTA, capable of computing GED in addition to traditional segmentation metrics, allowing comparison beyond Soft Dice performance. In addition to WBF, ABF and BCF fusions, we include two baseline methods. *EDL-MV* is an evidential network with a segmentation backbone producing 2 output channels instead of $2R$. During training, its GT is the majority voting over all annotations. *EDL-LS* is identical to EDL-MV in architecture, but the GT in each training iteration is a randomly sampled annotation. These serve as ablations, allowing to disentangle the contribution of using multiple per-rater segmentation heads and applying fusion.

Figure 2 shows probabilistic segmentation maps produced by our three TMS variants alongside the SOTA baselines. The second column visualizes the pixel-wise average of the annotations. In the easier task of optic disc segmentation, most methods align well with this averaged reference. For optic cup, TMS-WBF demonstrates strong uncertainty-awareness by assigning low probability to areas with high inter-rater disagreement. In contrast, TMS-BCF is overconfident in both tasks, which can be attributed to its ignoring of conflicts between sources. These patterns are also visually illustrated in Figure 3, which shows aleatoric and epistemic uncertainty maps for the three TMS variants. MRNet and TAB divert from the average GT especially around the optic cup boundaries.

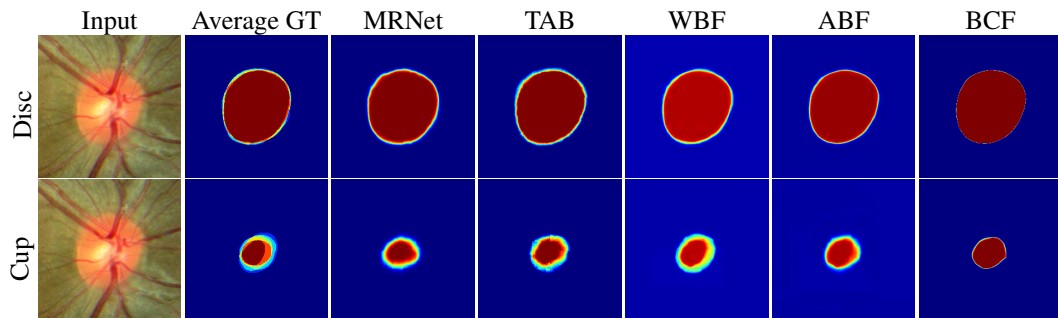

Figure 2: Comparison of optic disc (top row) and cup (bottom row) segmentation. Columns: input image, average GT, and predictions from MRNet, TAB, TMS-WBF, TMS-ABF, and TMS-BCF.

Table 1 shows the quantitative results on the RIGA dataset. Prediction-related measures are reported first, on the left. Our TMS variants outperform the SOTA baselines MRNet and TAB. In terms of GED and HM-IoU, comparisons between TMS variants are not meaningful, as for the calculation of these measures, sampling of predictions should be done prior to fusion. Uncertainty-related measures are reported afterwards, on the right. TMS-WBF achieves the best overall performance. For optic disc, it has the best NCC, AUCRef, and AURC, while for optic cup it remains highly competitive, closely matching TMS-ABF. Thus, despite the GT favoring ABF by giving equal weight to all raters, the uncertainty-aware weighting often yields superior results due to it reducing the impact of low-quality predictions.

The results show unintuitive patterns regarding the ECE score. TMS-BCF achieves low ECE scores even though it is overconfident and fails to capture disagreement regions. This can be explained by the way ECE is computed: each pixel contributes equally, and the score depends directly on the absolute predicted probabilities. Since the mistakes of overconfident methods are concentrated near object boundaries, which constitute a small fraction of the total pixels, their impact on the final score is minimal. In some cases, ECE is even lower on the easier optic disc task, which we attribute to the

Table 1: Quantitative results on the RIGA dataset. The best three performances per measure are in boldface, underlined, and italicized, respectively.

| Method | Prediction | | | Uncertainty | | | |
|---|---|---|---|---|---|---|---|
| | Soft Dice ↑ | HM-IoU ↑ (18) | GED ↓ (50) | NCC ↑ | AUCRef ↑ | AURC ↓ | ECE ↓ |
| **Disc** MRNet (Ji et al., 2021) | .977 ± .001 | - | - | - | - | - | - |
| TAB (Liao et al., 2023) | .977 ± .001 | - | .043 ± .000 | - | - | - | - |
| EDL-MV | .969 ± .000 | .939 ± .000 | .049 ± .000 | .630 ± .005 | .922 ± .001 | .026 ± .001 | .013 ± .000 |
| EDL-LS | .972 ± .000 | .938 ± .001 | .047 ± .001 | .685 ± .006 | .925 ± .000 | .023 ± .000 | .019 ± .002 |
| TMS-WBF | .978 ± .000 | .943 ± .000 | .029 ± .001 | .789 ± .002 | .930 ± .000 | .018 ± .000 | .046 ± .001 |
| TMS-ABF | .968 ± .015 | .933 ± .017 | .039 ± .018 | .766 ± .012 | .929 ± .000 | .019 ± .000 | .032 ± .013 |
| TMS-BCF | .968 ± .000 | .942 ± .001 | .029 ± .001 | .424 ± .010 | .921 ± .000 | .027 ± .000 | .012 ± .000 |
| **Cup** MRNet (Ji et al., 2021) | .859 ± .009 | - | - | - | - | - | - |
| TAB (Liao et al., 2023) | .872 ± .001 | - | .230 ± .005 | - | - | - | - |
| EDL-MV | .825 ± .003 | .778 ± .044 | .347 ± .061 | .434 ± .008 | .837 ± .001 | .113 ± .001 | .010 ± .001 |
| EDL-LS | .847 ± .002 | .736 ± .004 | .329 ± .157 | .577 ± .002 | .853 ± .001 | .097 ± .002 | .027 ± .007 |
| TMS-WBF | .884 ± .001 | .783 ± .002 | .115 ± .002 | .769 ± .004 | .873 ± .002 | .076 ± .002 | .041 ± .000 |
| TMS-ABF | .883 ± .003 | .782 ± .003 | .114 ± .006 | .765 ± .008 | .875 ± .001 | .075 ± .001 | .026 ± .007 |
| TMS-BCF | .844 ± .000 | .784 ± .001 | .112 ± .000 | .446 ± .004 | .847 ± .002 | .102 ± .002 | .013 ± .000 |

Table 2: Quantitative results on the LIDC dataset. The best three performances per measure are in boldface, underlined, and italicized, respectively. One NCC (*) is from the paper by Hu et al. (2019).

| Method | Prediction | | | Uncertainty | | | |
|---|---|---|---|---|---|---|---|
| | Soft Dice ↑ | HM-IoU ↑ (16) | GED ↓ (50) | NCC ↑ | AUCRef ↑ | AURC ↓ | ECE ↓ |
| Hu et al. (2019) | - | - | .280 ± .006 | .669 ± .011* | - | - | - |
| MoSE (Gao et al., 2023) | - | .574 ± .002 | .239 ± .002 | - | - | - | .001 ± .001 |
| EDL-MV | .633 ± .003 | .576 ± .003 | .448 ± .005 | .416 ± .012 | .603 ± .004 | .360 ± .004 | .002 ± .000 |
| EDL-LS | .656 ± .005 | .534 ± .007 | .443 ± .011 | .626 ± .001 | .661 ± .012 | .305 ± .019 | .002 ± .000 |
| TMS-WBF | .669 ± .002 | .683 ± .004 | .244 ± .003 | .680 ± .003 | .669 ± .007 | .295 ± .011 | .002 ± .001 |
| TMS-ABF | .710 ± .004 | .679 ± .005 | .248 ± .004 | .670 ± .003 | .697 ± .003 | .264 ± .004 | .001 ± .001 |
| TMS-BCF | .652 ± .005 | .677 ± .009 | .251 ± .007 | .434 ± .003 | .659 ± .008 | .301 ± .009 | .004 ± .004 |

larger foreground size. Overall, ECE does not adequately reflect model performance in this setting and should not be relied upon.

In contrast, NCC provides a more informative view: although still computed pixel-wise, it measures correlation, reducing sensitivity to absolute values of uncertainty estimates. As a result, TMS-WBF achieves the highest NCC scores across both tasks, while its ECE scores are not competitive due to producing probability estimates with higher entropy on average. Notably, NCC also reflects the largest performance gaps between methods, highlighting the importance of explicit uncertainty evaluation. For instance, TMS-BCF attains decent scores on other measures but exhibits a very low NCC score. However, due to its pixel-wise nature, NCC underestimates the difficulty gap between disc and cup segmentation, with TMS-BCF appearing stronger on cup. In contrast, AUCRef and AURC better reflect task difficulty and model performance, due to the aggregation of uncertainty estimates.

Figure 4 shows referral curves for WBF, ABF, and BCF. The $x$-axis represents the fraction of uncertain predictions that are rejected, while the $y$-axis shows Soft Dice computed over the remaining predictions. As increasingly uncertain predictions are removed, Soft Dice steadily improves, indicating that uncertainty estimates effectively identify less reliable predictions. Both WBF and ABF exhibit strong uncertainty-awareness. In contrast, BCF underperforms relative to both methods.

For the LIDC dataset, we compare against recent approaches that evaluate uncertainty at least to some extent. As discussed in Section 2.2, Hu et al. (2019) provide AU estimates and assess them using NCC. MoSE (Gao et al., 2023) reports calibration through the ECE score, though the authors acknowledge that ECE cannot capture multimodal or structural aspects of segmentation uncertainty and, therefore, treat it only as an auxiliary measure. As shown in Table 2, our TMS variants are the best across most measures, with the exception of GED, which is not a primary indicator in our

evaluation. TMS-ABF achieves the strongest performance, surpassing TMS-WBF on most measures. This is expected, since LIDC does not provide consistent annotator identities and thus limits WBF's ability to exploit rater-specific weighting. Nevertheless, the superior performance of TMS-ABF over EDL-MV and EDL-LS underlines the impact of multi-head modeling, which effectively leverages multiple predictions in an ensemble-like manner. Also, for both datasets, EDL-LS mostly outperforms EDL-MV, aligning well with the findings of prior work.

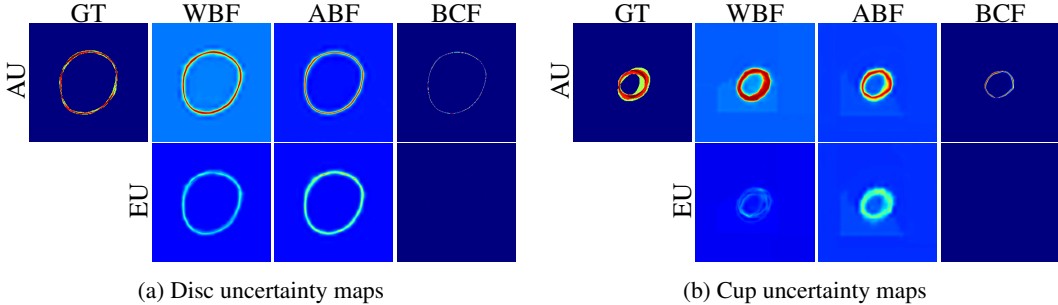

(a) Disc uncertainty maps          (b) Cup uncertainty maps

Figure 3: Uncertainty maps for the three TMS variants. The input image is the same as in Figure 2. Aleatoric and epistemic maps are max-normalized with values of 0.7 and 0.37, respectively.

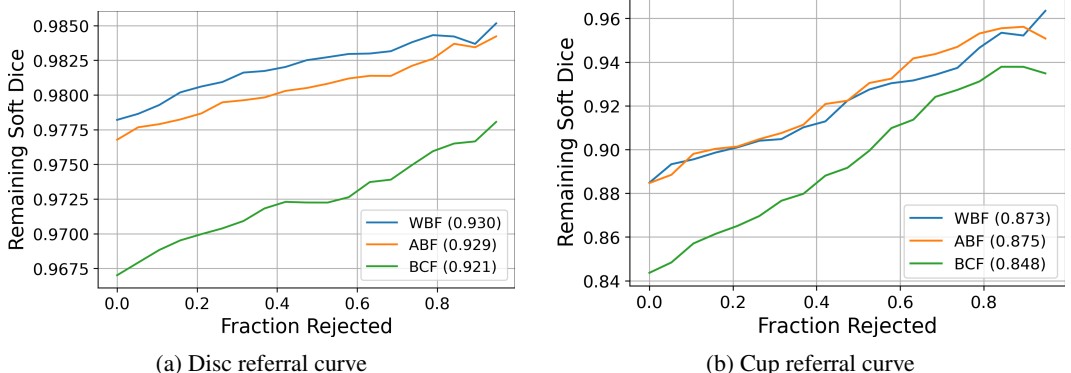

(a) Disc referral curve          (b) Cup referral curve

Figure 4: Referral curves illustrating the performance of uncertainty-based sample rejection. The area under the referral curve (AUCRef) is reported in parentheses.

# 7 CONCLUSIONS AND FUTURE WORK

We propose TMS, the first EDL method for multi-rater medical image segmentation. Our approach models network predictions associated with different raters as subjective opinions, which are then aggregated using WBF from SL. Beyond producing probabilistic segmentation maps, TMS provides explicit uncertainty estimates and effectively identifies potential low-quality segmentations. We further complement pixel-level estimates with aggregated image-level uncertainty scores. Across three segmentation tasks, our method achieves SOTA results in prediction-related measures and in AU estimation, the latter evidenced by superior NCC scores. In addition, we extend the evaluation protocol by incorporating AUCRef and AURC, allowing to evaluate the quality of overall predictive uncertainty estimates at image level.

In the future, we aim to design training and evaluation setups that account for varying rater reliability, allowing a more direct assessment of the benefits of weighted fusion. Another important direction is the direct evaluation of EU, for example through out-of-distribution detection. We also plan to study how the multi-rater setting influences the correlation between EU and AU, ultimately aiming to develop a more comprehensive framework for uncertainty estimation in multi-rater learning.

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

# A EVALUATION METHODS

## A.1 SOFT DICE

The Dice score is a widely used metric to evaluate segmentation performance, defined as

$$\text{Dice}(\hat{y}, y) = \frac{2|y \cap \hat{y}|}{|y| + |\hat{y}|}, \tag{12}$$

where $\hat{y}$ and $y$ represent the prediction mask and the GT mask, respectively. However, we are interested not only in the final binary predictions but also in how well the predicted probabilities reflect the agreement between experts. To this end, we employ the Soft Dice score. It is computed by evaluating the predictions across multiple probability thresholds, specifically $0.1, 0.3, 0.5, 0.7, 0.9$. At each threshold, both the predicted probability map and the soft GT map—obtained by averaging the annotations from all raters—are binarized, and the Dice score is computed. The Soft Dice score is then obtained by averaging the Dice scores across all thresholds.

## A.2 GENERALIZED ENERGY DISTANCE

As described by Kahl et al. (2024), the Generalized Energy Distance (GED) is defined as

$$D_{\text{GED}}^2(p, \hat{p}) = 2\, \mathbb{E}_{y \sim p,\, \hat{y} \sim \hat{p}}[d(y, \hat{y})] - \mathbb{E}_{y, y' \sim p}[d(y, y')] - \mathbb{E}_{\hat{y}, \hat{y}' \sim \hat{p}}[d(\hat{y}, \hat{y}')]. \tag{13}$$

Here, $p$ denotes the distribution of reference annotations, and $\hat{p}$ is the predicted distribution of segmentation masks; $d(y, \hat{y})$ denotes the distance between a reference annotation $y$ and a generated sample $\hat{y}$, while $d(y, y')$ and $d(\hat{y}, \hat{y}')$ measure the distances between pairs of annotations and pairs of generated samples, respectively. For the distance, we use the complement of the Intersection over Union (IoU):

$$d(a, b) = 1 - \text{IoU}(a, b). \tag{14}$$

As mentioned by Hu et al. (2019), this metric allows to simultaneously capture accuracy and diversity. However, as noted by Kohl et al. (2019), GED rewards sample diversity and may yield low values even when the generated distribution does not match the ground truth well.

In our TMS framework, predictions are sampled *before* the fusion stage. To obtain $P$ predictions from $R$ raters, we draw $\lfloor P/R \rfloor$ samples from each rater and distribute the remaining $P \bmod R$ samples across raters. Since the reference annotations are not fused either, this ensures that GED evaluates both the diversity of the predictions and their distributional match to the individual references.

## A.3 HUNGARIAN-MATCHED IOU

To address the limitation of GED noted in the previous subsection, we additionally compute the Hungarian-Matched IoU (HM-IoU) following Kohl et al. (2019). This metric employs the Hungarian algorithm (Kuhn, 1955; Munkres, 1957) to find an optimal one-to-one matching between model predictions and reference annotations, using IoU as the similarity measure. To ensure equal set sizes, we repeat the GT samples until their number matches the number of generated predictions. The final HM-IoU score is obtained as the average IoU over all matched pairs. Similar to GED, we sample the predictions before fusion.

## A.4 NORMALIZED CROSS-CORRELATION

To evaluate whether the predicted aleatoric uncertainties reflect the level of agreement between different raters, we use the Normalized Cross-Correlation (NCC), as described by Kahl et al. (2024):

$$\text{NCC} = \frac{1}{N \sigma_a \sigma_b} \sum_{i=1}^{N} (a_i - \mu_a) \times (b_i - \mu_b), \tag{15}$$

where $a$ and $b$ denote the reference and predicted uncertainty maps, respectively, and $N$ is the number of pixels. The terms $\mu_a$ and $\mu_b$ are the means, and $\sigma_a$ and $\sigma_b$ are the standard deviations of the respective maps. The reference uncertainty map $a$ is constructed by calculating the pixel-wise variance across $R$ different segmentation annotations.

### A.5 AREA UNDER THE REFERRAL CURVE

To assess the effectiveness of the uncertainty estimates in identifying low-quality segmentations, we adopt a referral-based mechanism as described by Lambert et al. (2024). In this setting, we aggregate the pixel-level uncertainty estimates to obtain a single uncertainty estimate for a given image, using the patch-level aggregation mechanism described in Section 3.2. Since we want to detect low-quality segmentations regardless of the source of uncertainty, we compute the score by summing AU and EU at the pixel level before aggregation. This aggregated score allows us to sort the predictions from least to most certain. We then progressively remove (i.e., refer to an expert) a fraction of the most uncertain samples, and compute the Soft Dice score on the remaining set. This yields a referral curve that shows how segmentation quality varies as increasingly uncertain predictions are excluded. The Area under the Referral Curve (AUCRef) is used as a qualitative score. A higher AUCRef indicates that rejecting uncertain samples results in improved segmentation quality, suggesting that the uncertainty estimates are effective at identifying potentially erroneous predictions.

### A.6 AREA UNDER THE RISK-COVERAGE CURVE

The Area under the Risk-Coverage Curve (AURC) is a metric used in selective classification to evaluate how well a system balances minimizing risk (i.e., reducing prediction errors) while maximizing coverage (i.e., minimizing the number of cases left out for manual correction). To compute AURC in the context of semantic segmentation, we follow the description provided by Kahl et al. (2024). Given an evaluation dataset $D = \{(x_i, y_i)\}_{i=1}^n$ and a predictor $f$, the confidence scoring function $g(x_i)$ is defined as the negative uncertainty score, and the risk associated with a prediction is computed as

$$l(x, y, f) = 1 - \text{Dice}(f(x), y). \tag{16}$$

Given a confidence threshold $\tau$, the selective risk is computed as

$$\text{Risk}(\tau \mid f, g, D) = \frac{\sum_{i=1}^n l(x_i, y_i, f) \times \mathbb{I}(g(x_i) \geq \tau)}{\sum_{i=1}^n \mathbb{I}(g(x_i) \geq \tau)}, \tag{17}$$

and the coverage is defined as

$$\text{Coverage}(\tau \mid g, D) = \frac{\sum_{i=1}^n \mathbb{I}(g(x_i) \geq \tau)}{n}. \tag{18}$$

The AURC over a sorted list of thresholds $\{\tau_t\}_{t=1}^T$ is then computed as

$$\text{AURC}(f, g, D) = \sum_{t=1}^T \left(\text{Coverage}(\tau_t) - \text{Coverage}(\tau_{t-1})\right) \times \frac{\text{Risk}(\tau_t) + \text{Risk}(\tau_{t-1})}{2}, \tag{19}$$

where conditioning on $f$, $g$, and $D$ on the right-hand side is omitted for clarity.

### A.7 EXPECTED CALIBRATION ERROR

Following the description given by Gao et al. (2023), we evaluate the difference between predicted probabilities and the actual accuracy using the expected calibration error (ECE), defined as

$$\text{ECE} = \mathbb{E}_{\hat{P}} \left[ \left| P(\hat{Y} = Y \mid \hat{Y} = p) - p \right| \right]. \tag{20}$$

Here, $Y$ denotes random variable for the GT label, $\hat{Y}$ denotes the predicted class, and $\hat{P}$ is the associated predicted probability. The pixel-wise label distribution and the predictive distribution are computed by marginalization and treating each pixel as independent and identically distributed (IID).

We note that while Gao et al. (2023) compute ECE using 16 sampled predictions, we instead use the probabilities obtained from Equation 3. This choice reflects the objective of our method, which is to provide probabilistic segmentation maps together with uncertainty maps, rather than to generate diverse predictions, and therefore offers a more faithful assessment of our model's calibration quality.

## B  QUALITATIVE RESULTS

Figure 5 shows probabilistic segmentation maps produced by our three TMS variants alongside the SOTA baselines, for a sample exhibiting significant inter-rater variability. Figure 6 illustrates aleatoric and epistemic uncertainty maps produced by the three TMS variants for the same input image.

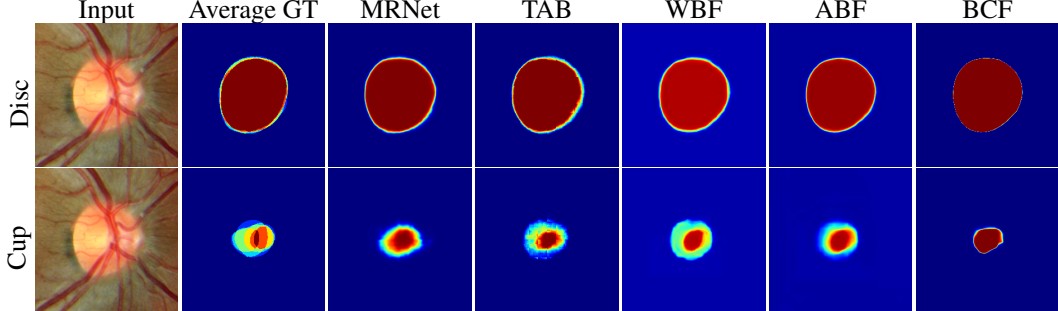

Figure 5: Comparison of optic disc (top row) and cup (bottom row) segmentation for a sample with significant inter-rater variability. Columns: input image, average GT, and predictions from MRNet, TAB, TMS-WBF, TMS-ABF, and TMS-BCF.

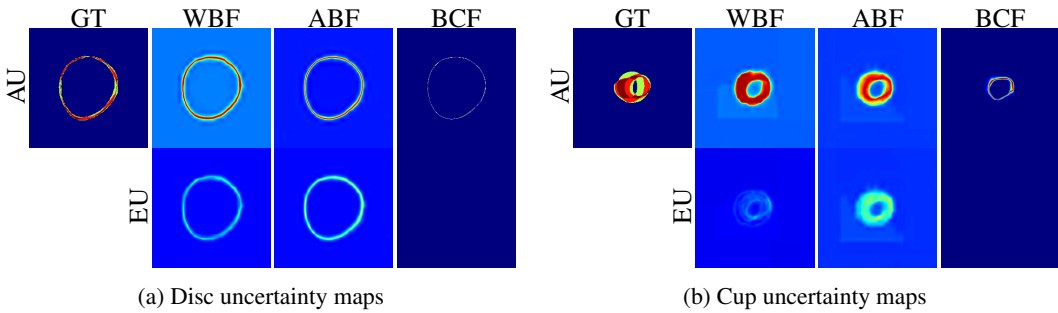

(a) Disc uncertainty maps                    (b) Cup uncertainty maps

Figure 6: Uncertainty maps for the three TMS variants. The input image is the same as in Figure 5. Aleatoric and epistemic maps are max-normalized with values of 0.7 and 0.37, respectively.

## C  DATASET DETAILS

The RIGA benchmark (Almazroa et al., 2017) is a publicly available dataset for retinal optic cup and disc segmentation, comprising a total of 750 color fundus images from three sources: 460 images from MESSIDOR, 195 images from BinRushed, and 95 from Magrabia. This segmentation task is inherently multi-label, as the optic cup is a subset of the optic disc. To address this, we treat it as two separate binary segmentation tasks. Each image in RIGA is manually annotated by six glaucoma experts from different institutions. For model training, we follow the setup of Ji et al. (2021); Liao et al. (2023), using the combined 195 BinRushed and 460 MESSIDOR images as the training set, while reserving the 95 images from Magrabia as a test set. From the training set, we randomly allocate 15% as a validation set, ensuring proportional sampling from both MESSIDOR and BinRushed. It should be noted that we use the version of the dataset from Ji et al. (2021) that is cropped around the foreground, and resize all images to $256 \times 256$ pixels. Input normalization is applied using RGB mean values of $[0.485, 0.456, 0.406]$ and standard deviations of $[0.229, 0.224, 0.225]$.

The LIDC dataset (Armato III et al., 2011) contains 1018 3D thoracic CT scans. Each image is annotated by 4 out of a pool of 12 annotators. Despite the variability in annotator identity, LIDC is one of the largest and most widely used datasets for multi-rater segmentation. Several recent works—particularly those focused on modeling uncertainty—have evaluated on LIDC, and we include it to enable meaningful comparisons with these approaches. We use the preprocessed 2D version of LIDC released by Kohl et al. (2018), which consists of 15,096 slices cropped to $128 \times 128$ pixels

and centered on lesion regions. Following the protocol of Hu et al. (2019), we split the data into training, validation, and test sets using a 70%/15%/15% ratio.

# D    EXPERIMENTAL SETUP

For the segmentation backbone, we employ DeepLabV3 (Chen et al., 2017) with a ResNet-101 backbone. The model is initialized with weights pretrained on a subset of the COCO dataset (Lin et al., 2014), restricted to the 20 categories overlapping with Pascal VOC (Everingham et al., 2010). Training is performed using the Adam optimizer (Kingma & Ba, 2015), with 200 epochs for RIGA and 100 epochs for LIDC. Among the trained checkpoints, the model achieving the lowest validation loss is selected for evaluation.

Learning rate scheduling is managed by PyTorch's `ReduceLROnPlateau` scheduler with a decay factor of 0.5, a patience of four epochs, and a lower bound of $1 \times 10^{-6}$. The initial learning rate is set to 0.0001 for RIGA and 0.00005 for LIDC. For the uncertainty aggregation step, the patch side length is chosen as 65 pixels on RIGA and 5 pixels on LIDC. A batch size of 16 is used for all experiments. For the WBF method, hyperparameters are tuned on the validation set, and the optimal configuration is fixed for all subsequent variants of our method.

For the SOTA baselines, we deliberately select approaches that provide official PyTorch implementations and evaluate on the datasets originally used in their respective papers. Consequently, we retain all hyperparameters as specified in the released code. We note that the MoSE implementation computes GED with 48 samples instead of 50, reflecting an implementation-specific detail.

All experiments are conducted on a single NVIDIA GeForce RTX 4080 GPU with 16 GB memory. The CPU is an Intel Core i7-13700F with 16 GB RAM. To ensure robustness, each experiment is repeated three times with different random seeds, and we report the mean and standard deviation across these runs.

