# OpenReview forum: "Trusted Multi-Rater Segmentation"
_ICLR.cc/2026/Conference — ICLR 2026 Conference Withdrawn Submission_

### Official Review · Reviewer_2rNM · 2025-10-25

**Soundness:** 3
**Presentation:** 3
**Contribution:** 2
**Rating:** 6
**Confidence:** 4

**Summary:**

This paper presents Trusted Multi-Rater Segmentation (TMS), a framework that integrates Evidential Deep Learning (EDL) and Subjective Logic (SL) to jointly model inter-rater variability and estimate epistemic and aleatoric uncertainty in medical image segmentation. The method treats each annotator’s prediction as a subjective opinion parameterized by a Dirichlet distribution and fuses them through weighted belief fusion (WBF) to obtain uncertainty-aware segmentation maps. Experiments on RIGA and LIDC datasets demonstrate improved segmentation accuracy and uncertainty calibration over existing baselines such as MRNet and TAB.

The paper is well organized, technically sound, and addresses a relevant topic—trustworthy uncertainty modeling under inter-rater variability. However, the methodological innovation is moderate, and comparisons against several recent, stronger multi-rater probabilistic models are missing. Overall, the contribution is conceptually clear but not yet fully validated experimentally.

**Strengths:**

1. The work tackles uncertainty estimation under label disagreement, a crucial yet underexplored issue in medical AI and trustworthy segmentation.

2. The integration of EDL with subjective logic provides an intuitive framework for combining rater-specific uncertainty into a fused belief distribution.

3. The authors evaluate not only segmentation quality (Soft Dice, HM-IoU) but also multiple uncertainty metrics (NCC, AUCRef, AURC), which enrich the analysis.

4. TMS shows consistent performance improvement over MRNet and TAB, with notably stronger uncertainty-awareness.

5. The paper is clearly structured, mathematically detailed, and well motivated.

**Weaknesses:**

1. Novelty. The core contribution lies in combining existing frameworks (Evidential Deep Learning + Weighted Belief Fusion) rather than introducing fundamentally new architectures or learning objectives. The fusion rule and uncertainty computation directly follow established formulations in subjective logic.

2. Lack of comparison with latest probabilistic multi-rater models. The paper omits comparison with several recent state-of-the-art approaches that explicitly address inter- and intra-observer variability, such as: Probabilistic modeling of inter- and intra-observer variability in medical image segmentation (ICCV 2023), Diversified and Personalized Multi-Rater Medical Image Segmentation (CVPR 2024), both of which propose more expressive probabilistic or personalized frameworks for capturing annotator heterogeneity. Without such baselines, it is unclear whether TMS provides substantive advantages over probabilistic or generative formulations.

3. The evaluation is limited to two small-scale datasets (RIGA, LIDC). There is no validation on broader benchmarks such as QUBIQ Challenge (Li, Hongwei Bran, et al. "Qubiq: Uncertainty quantification for biomedical image segmentation challenge"), NPC Dataset (Diversified and Personalized Multi-Rater Medical Image Segmentation), SUN-SEG Dataset (Ji et al., 2022, Video polyp segmentation: A deep learning perspective). Inclusion of these datasets would provide stronger evidence of robustness and generalizability across modalities and rater configurations.

4. The proposed model assumes consistent annotators across images, which holds for RIGA but not for LIDC. On datasets with varying annotator participation, the WBF weighting mechanism becomes ill-defined, undermining the claimed interpretability and limiting broader applicability.

5. The paper does not analyze how uncertainty weighting affects fusion results or calibration quality. There is no visualization or quantitative correlation between uncertainty weights and annotator reliability, which weakens the interpretability claim.

6. The quantitative gains (e.g., <1% in Soft Dice) are small relative to the added model complexity, and the authors’ claim of “state-of-the-art” performance may be overstated given missing baselines and narrow evaluation scope.

**Questions:**

See weaknesses.

---

### Official Review · Reviewer_mdwL · 2025-10-26

**Soundness:** 2
**Presentation:** 2
**Contribution:** 2
**Rating:** 4
**Confidence:** 4

**Summary:**

This paper presents a new approach to multi-rater medical image segmentation that integrates evidential deep learning with subjective logic for uncertainty quantification. Although effective, this article still has some deficiencies in terms of innovation, experiment comparisons and practical application, and has not yet reached the acceptance bar of ICLR.

**Strengths:**

1.	First application of evidential deep learning in multi-rater medical image segmentation, combining EDL with subjective logic.

2.	Experimental results demonstrate state-of-the-art performance across multiple metrics and datasets.

**Weaknesses:**

1. The model seems too easy. This paper mainly introduces evidential deep learning into multi-rater segmentation, which is not novel enough in my view.
2. This paper only tested on two datasets (RIGA and LIDC), which potentially limits generalizability.
3. The multi-head architecture with 2R output channels may be computationally expensive for large numbers of raters.
4. Recent multi-rater methods not included in comparison[1,2].

     [1] Diversified and personalized multi-rater medical image segmentation. CVPR2024.

     [2] Multi-rater Prompting for Ambiguous Medical Image Segmentation. BIBM2024.

**Questions:**

1. How does the method scale with increasing numbers of raters (beyond 6-12)? What is the computational cost?

2. Have the uncertainty maps been evaluated by medical experts for clinical relevance and interpretability?

---

### Official Review · Reviewer_5QKX · 2025-10-31

**Soundness:** 2
**Presentation:** 2
**Contribution:** 2
**Rating:** 2
**Confidence:** 5

**Summary:**

In this manuscript, the authors propose Trusted Multi-Rater Segmentation (TMS), a segmentation algorithm utilizing evidential deep learning (EDL) for multi-rater medical segmentation tasks. A standard encoder-decoder segmentation backbone is utilized to generate predictions corresponding to different annotators, and a weighted belief fusion (WBF) method is adopted to fuse prediction for the final result. The method presents significantly better performance compared with some previous multi-rater segmentation methods.

**Strengths:**

1. This study is the first to utilize EDL for multi-rater medical segmentation tasks.
2. The proposed method is validated with two datasets of different modalities to prove its generalizability.

**Weaknesses:**

1. **Lack of Novelty:**  The methodological section demonstrates limited innovation. The overall framework appears to be an assemblage of existing components. Specifically, the network backbone employs the DeepLabV3 architecture, the calculation of the Dirichlet distribution and its associated loss functions is directly adopted from (Han et al., 2021), and the multi-rater fusion strategies tested—namely WBF, ABF, and BCF—are all established methods from the literature.
2. **Insufficient Implementation Details:** The description of the methodology lacks critical details, which hinders reproducibility. Specific omissions will be outlined in the 'Questions' section. Furthermore, the loss function coefficients $\lambda_t$, $\lambda_p$, and $\lambda_s$ are assigned directly without justification or supporting ablation studies to analyze their impact or validate the chosen values.
3. **Inadequate Experimental Comparison:** The experimental evaluation is not fully comprehensive. The comparison with recent methods (e.g., [MrPrism](https://doi.org/10.1016/j.scib.2024.06.037)) specifically designed for multi-rater medical segmentation is missing. Additionally, the validation is conducted on a relatively limited set of datasets. Notably, the MRNet study utilized the QUBIC dataset, which is designed for multi-organ segmentation. The absence of experiments on this or similar datasets limits the assessment of the proposed method's generalizability.

**Questions:**

1. In Figure 1, does $Evidence_r$ represent $e^r_1, ..., e^r_k$? If the answer is yes, why do you not illustrate it in the figure or text? I cannot find the expression $Evidence_r$ otherwise. So does  $Dirichlet_r$.
2. In Figure 1, what do the terms "Combined Dirichlet", "Prediction", and "Uncertainty" represent? I cannot find the corresponding mathematical expressions.
3. How to calculate $u_r$ in Eq. 7?
4. According to Eq. 7, the fusion method is applied to the evidence $e^r_k$. However, the fusion method is applied to Dirichlet distribution in Figure 1.
5. What are the values of $\alpha$ and $\beta$ in Eq. 9?
6. Only results of the RIGA dataset are visually compared. How about the LIDC dataset?
7. According to Table 1 and Table 2, TMS-WBF does not present superiority compared with TMS-ABF. Why WBF is adopted in the final version?

---

### Official Review · Reviewer_sxvi · 2025-11-01

**Soundness:** 3
**Presentation:** 2
**Contribution:** 3
**Rating:** 4
**Confidence:** 4

**Summary:**

The paper proposes,TMS, a multi‑rater medical image segmentation framework that uses evidential deep learning (EDL) to output Dirichlet parameters per rater and then combines these rater‑specific “subjective opinions” with **weighted belief fusion (WBF)** from subjective logic. Concretely, the network predicts 2R channels (foreground/background for each of the $\(R\)$ raters), converts logits to non‑negative **evidence** via Softplus, forms per‑rater Dirichlet distributions $\(\alpha_k=e_k+1\)$, and fuses them using a confidence‑weighted rule $\(e_k=\tfrac{\sum_r e_k^{(r)}(1-u^{(r)})}{\sum_r(1-u^{(r)})}\)$ (Eq. 7). The loss per rater is the EDL expected cross‑entropy plus a KL regularizer toward a uniform Dirichlet, augmented with a Dice term, and the overall loss sums over raters (Eqs. 4–6). TMS outputs probabilistic segmentations and separates **epistemic** (via $\(K/S\)$) and **aleatoric** (expected entropy) uncertainty in one forward pass.

**Strengths:**

1. Casting each rater’s output as a Dirichlet opinion makes both belief and uncertainty explicit (Eqs. 1–3), with EU/AU derived in closed form.
2. Using WBF weights opinions by confidence \((1-u)\), mitigating low‑quality/uncertain raters; the paper also fairly compares ABF and BCF and explains when each helps.
3. The work reports NCC for AU‑vs‑variance alignment and image‑level AUCRef/AURC using a patch‑aggregation strategy to reduce object‑size confounds.
4. EDL yields EU/AU without ensembles or MC sampling, keeping inference deterministic and fast; training/inference setups and schedules are well documented.
5. On RIGA, TMS‑WBF leads Soft‑Dice and uncertainty metrics; on LIDC, TMS‑ABF is strongest, with an honest discussion that identity inconsistency reduces WBF’s advantage.

**Weaknesses:**

1. The approach **assumes the same set of annotators per image** (Sec. 3.2), yet **LIDC** violates this; indeed ABF outperforms WBF there (Table 2). This weakens the case for WBF in common settings where rater identities vary or are unknown.
2. The head outputs **\(2R\)** channels (foreground/background per rater). Extending to multi‑class problems implies **\(KR\)** channels, which may be computational expensive when \(R\) or \(K\) is large.
3. EU is computed as \(K/S\) from a single deterministic model (Sec. 3.1). While standard in EDL, this **does not capture parameter uncertainty** as in Bayesian/ensemble methods; thus “EU” reflects “lack of evidence” rather than true model‑parameter uncertainty. The paper acknowledges future EU evaluation only briefly.
4.  A rater that is **confident but wrong** would be up‑weighted; there is no learned calibration of weights to rater quality beyond uncertainty. Empirical safeguards or learned fusion could strengthen robustness.
5. For GED/HM‑IoU, predictions are **sampled from per‑pixel Dirichlet** before fusion (App. A.2). Pixel‑wise independence can yield spatially incoherent samples, potentially underestimating the benefit of spatial generative models. The paper does not analyze this effect.
6. The paper evaluates AU (NCC) and overall uncertainty quality (AUCRef/AURC) but **does not directly test EU** (e.g., OOD detection, shift robustness). This leaves the EU claim only partially validated.
7. For LIDC, where each slice is annotated by a **subset** and identities vary, it is unclear how the **2R per‑rater outputs** are consistently mapped across images during training, beyond noting why ABF works better.
8. The paper frames outputs as **foreground/background** per rater (Sec. 3.2). It would help to show multi‑class experiments or a discussion of memory/time when moving to \(K>2\).

**Questions:**

1. How are the **2R rater‑specific channels** trained for **LIDC**, where annotator identities change between images (Sec. 4)? Is there a fixed slotting or random assignment, and how might that affect WBF/ABF?
2. Do you observe cases where WBF **over‑trusts** a rater with low \(u\) but systematic bias? Qualitative examples (like those in Figs. 2–3) highlighting fusion success/failure would be instructive.
3. WBF uses weights \(1-u\) derived from the Dirichlet strength (Eq. 7). Have you tried **learning** rater weights (e.g., a small gating network) or calibrating \(u\) against validation performance to avoid up‑weighting confident‑but‑wrong opinions?
4. Beyond AUCRef/AURC/NCC, can you report **EU‑specific** tests (e.g., OOD detection, covariate/label‑shift experiments) to demonstrate that the \(K/S\) signal tracks epistemic risk?
5.  When sampling pre‑fusion predictions for **GED/HM‑IoU** (App. A.2), are probabilities sampled **independently per pixel**? If so, have you explored spatial regularization (e.g., CRF post‑processing) or correlated sampling to improve sample realism?
6.  How sensitive are results to $\(\lambda_t,\lambda_s,\lambda_p\)$ (Eq. 10) and to the **patch size** used for image‑level uncertainty aggregation (65 vs 5; App. D)? Could you provide a brief study or heuristic for choosing patch size by task?
7. What are the memory/time trade‑offs for moving from binary to **multi‑class** segmentation (\(KR\) outputs) or to larger \(R\) (e.g., crowd‑sourcing with tens of annotators)? Any parameter‑sharing schemes under consideration?

---

### Note · Authors · 2025-12-01

I have read and agree with the venue's withdrawal policy on behalf of myself and my co-authors.